# Associations between low body mass index and mortality in patients with sepsis: A retrospective analysis of a cohort study in Japan

Tetsuya Sato[1]*, Daisuke Kudo[1,2], Shigeki Kushimoto[1,2], Masatsugu Hasegawa[3], Fumihito Ito[4], Sathoshi Yamanouchi[5], Hiroyuki Honda[6], Kohkichi Andoh[5], Hajime Furukawa[1], Yasuo Yamada[7], Yuta Tsujimoto[8], Manabu Okuyama[9], Masakazu Kobayashi[1]

1 Department of Emergency and Critical Care Medicine, Tohoku University Hospital, Sendai, Japan,
2 Division of Emergency and Critical Care Medicine, Tohoku University Graduate School of Medicine, Sendai, Japan, 3 Department of Emergency and Critical Care Medicine, Japanese Red Cross Ishinomaki Hospital, Ishinomaki, Japan, 4 Department of Emergency Medicine, International University of Health and Welfare, Narita, Japan, 5 Emergency and Critical Care Department, Sendai City Hospital, Sendai, Japan, 6 Department of Advanced Disaster Medicine and Emergency Critical Care Center, Niigata University Medical and Dental Hospital, Niigata, Japan, 7 Department of Emergency Medicine, Sendai Medical Center, Sendai, Japan, 8 Department of Emergency and Critical Care Medicine, Yamagata Prefectural Central Hospital, Yamagata, Japan, 9 Department of Critical Care Medicine, Akita University Hospital, Akita, Japan

* t23jor@med.tohoku.ac.jp

**Data Availability Statement:** The datasets generated and analyzed during the current study are publicly available at https://data.mendeley.com/datasets/vvv89kw3k5/1.

## Abstract

### Background

The distribution of body mass in populations of Western countries differs from that of populations of East Asian countries. In East Asian countries, fewer people have a high body mass index than those in Western countries. In Japan, the country with the highest number of older adults worldwide, many people have a low body mass index. Therefore, this study aimed to determine the association between a low body mass index and mortality in patients with sepsis in Japan.

### Methods

We conducted this retrospective analysis of 548 patients with severe sepsis from a multicenter prospective observational study. Multivariate logistic regression analyses determined the association between body mass index and 28-day mortality adjusted for age, sex, pre-existing conditions, the occurrence of septic shock, Acute Physiology and Chronic Health Evaluation II scores, and Sequential Organ Failure Assessment scores. Furthermore, the association between a low body mass index and 28-day mortality was analyzed.

### Results

The low body mass index group represented 18.8% of the study population (103/548); the normal body mass index group, 57.3% (314/548); and the high body mass index group, 23.9% (131/548), with the 28-day mortality rates being 21.4% (22/103), 11.2% (35/314),

**Funding:** This work was supported by JSPS KAKENHI Grant Number JP19H03755.

**Competing interests:** The authors have declared that no competing interests exist.

and 14.5% (19/131), respectively. In the low body mass index group, the crude and adjusted odds ratios (95% confidence intervals) for 28-day mortality relative to the non-low body mass index (normal and high body mass index groups combined) group were 2.0 (1.1–3.4) and 2.3 (1.2–4.2), respectively.

## Conclusion

A low body mass index was found to be associated with a higher 28-day mortality than the non-low body mass index in patients with sepsis in Japan. Given that older adults often have a low body mass index, these patients should be monitored closely to reduce the occurrence of negative outcomes.

## Introduction

Sepsis is a complex syndrome characterized by physiological, pathological, and biochemical abnormalities caused by severe infection [1]. The effects of sepsis on adipose tissue have been reported in several studies as one of the complicating pathophysiological factors [2–6]. Adipose tissue plays an important role in homeostasis, secreting adipokines in response to various signals, controlling feeding, thermoregulation, immunity, and neuroendocrine functions [4]. There is no generally accepted device to measure the amount and function of adipose tissue in clinical settings. Therefore, instead of direct measurements of adipose tissue, the body mass index (BMI) has been used as an alternative indicator, because the amount of adipose tissue generally increases with increasing BMI [7–9].

The obesity paradox refers to the hypothesis that being obese may lead to better outcomes than being normal weight [2]. Obesity increases the risk of obesity-related chronic diseases but is paradoxically associated with increased survival in patients with acute conditions, including critical illness [10], especially in older adults [11–13]. Recent studies from Western countries have shown that a high BMI is associated with reduced mortality in patients with sepsis [14–16]. In these studies, only a few patients with a low BMI were included, and the association between a low BMI and the outcomes of patients with sepsis was not clarified. However, a recent Chinese report on patients with sepsis suggested that a low BMI was associated with high mortality rates [17]. In East Asian countries, fewer people have a high BMI than in Western countries [18]. In general, Asians have a lower BMI than people of European descent, with differences ranging from –0.3 to –3.6 kg/m$^2$ [19]. In Japan, the country with the highest number of older adults worldwide [20], many people have a low BMI [21]. Thus, research focusing on the influence of a low BMI in patients with sepsis is warranted, especially in East Asian countries.

Therefore, the current study aimed to elucidate the association between low BMI and mortality in patients with sepsis.

## Materials and methods

### Ethics statements

This study was approved by the Ethics Review Board of the Tohoku University Graduate School of Medicine based on the guidelines of the Ethics Committee of the Graduate School of Medicine, Tohoku University (No. 2013-1-42). The Institutional Review Boards (IRB) of each participating institution also approved the study. Research was conducted according to the Declaration of Helsinki. The requirement for informed consent was waived by all IRBs owing

to the observational nature of the study and the lack of treatment beyond that performed as part of daily clinical practice, in accordance with Japanese Guideline (Ministry of Education, Culture, Sports, Science and Technology, and Ministry of Health, Labor, and Welfare, Japan. Ethical Guidelines for Medical and Health Research Involving Human Subjects, March 2015). All data were fully anonymized prior to access, which occurred from January 2015 to September 2016.

## Study design

This study used data from the Tohoku Sepsis Registry (University Hospital Medical Information Network Clinical Trials Registry: UMIN000010297), a multicenter observational cohort study conducted at 10 institutions—including three university hospitals and seven community hospitals—in the Tohoku District in northern Japan.

The detailed design of this study has been reported previously [22]. Briefly, the Tohoku Sepsis Registry prospectively enrolled consecutive patients admitted to an intensive care unit (ICU) with severe sepsis or patients who developed severe sepsis after admission to an ICU between January 2015 and December 2015. Severe sepsis and septic shock were defined according to the 2012 International Sepsis Guidelines [23]. Patients aged <18 years were excluded from the registry. Researchers at each institution collected data from patient medical records and registered them in the web registration system. The data used in the current analysis included age, sex, BMI upon admission, pre-existing conditions (cardiovascular disease, stroke, chronic obstructive pulmonary disease, autoimmune disease, chronic liver disease, diabetes mellitus, chronic kidney disease, or malignancy), medication before admission, lactate levels, occurrence of septic shock, Acute Physiology and Chronic Health Evaluation (APACHE) II scores [24], Sequential Organ Failure Assessment (SOFA) scores upon admission [25], duration of ICU stay, and 28-day and in-hospital mortality.

## Definitions and outcome measures

BMI was defined as the weight in kilograms divided by the square of the height in meters. Patients were divided into the following three groups according to their BMI on admission: low BMI ($<18.5 \text{ kg/m}^2$), normal BMI ($18.5 \leq \text{BMI} < 25.0 \text{ kg/m}^2$), and high BMI ($\geq 25.0 \text{ kg/m}^2$) according to the 2016 Japanese Guidelines for the Management of Obesity Disease [16, 26]. Additionally, to examine the influence of a low BMI on mortality, the normal and high BMI groups were combined into a non-low BMI group for comparison with the low BMI group.

The primary outcome measure was all-cause 28-day mortality. No follow-up after discharge, survival discharge, or survival during hospitalization within 28 days was regarded as survival, and in-hospital death within 28 days was regarded as death. In survival time analysis, survival was measured as number of days from admit date to death and censor was defined as survival discharge within 28 days. Secondary outcomes were all-cause in-hospital mortality and ICU-free days. The number of ICU-free days within a 28-day period was calculated by subtracting the duration of ICU stay from 28 days. If a patient died before discharge from the ICU, then the number of ICU-free days was calculated as zero.

## Statistical analyses

Categorical and continuous variables are expressed as numbers (%) or medians (interquartile range). Imputation methods were not used to complete the dataset for any missing values. Values were compared using Pearson's chi-square test for categorical variables, the Mann–Whitney $U$-test for two-group comparisons of continuous variables, and the Kruskal–Wallis test for three-group comparisons. Independent factors associated with mortality were examined using

multivariate logistic regression analysis and Cox proportional hazards regression models, respectively. In the regression models, the following variables were used for adjustment: age, sex, presence of pre-existing conditions (cardiovascular disease, stroke, chronic obstructive pulmonary disease, autoimmune disease, chronic liver disease, diabetes mellitus, chronic kidney disease, or malignancy), occurrence of septic shock, APACHE II and SOFA scores, and lactate levels. We deemed these variables potentially important based on clinical judgment and past sepsis research [27]. A previous study also indicated that body weight may be associated with age and pre-existing conditions [28]. Co-linearity between variables was excluded before modeling by determining the correlation coefficient. We decided on the number of variables to be adjusted based on the numbers of non-survivors on day 28. This study was a secondary analysis that was not planned in advance, and there was no preset sample size. Data were analyzed using JMP Pro (version 13.0) software (SAS Institute Japan Ltd., Tokyo, Japan). All statistical tests were two sided, and p-values <0.05 for single comparisons and <0.016 for multiple comparisons (after Bonferroni correction) were considered statistically significant.

## Results

A total of 616 patients were registered in the Tohoku Sepsis Registry. Of those, 43 patients were withdrawn from the aggressive treatment phase within 4 days of the diagnosis of severe sepsis and were thus excluded. Another 25 patients were excluded owing to the lack of BMI information. Finally, the data of 548 patients were analyzed in the current study (**Fig 1**).

### Demographics and clinical characteristics

The median (interquartile range) age of the study population was 74.5 (64–83) years, and 63% (345/548) of the patients were male. The median (interquartile range) BMI was 22 (19.3–24.8) kg/m$^2$. The patient demographics and severity on admission in each group are shown in **Table 1**. There were 18.8% (103/548) of patients with a low BMI, 57.3% (314/548) with a normal BMI, and 23.9% (131/548) with a high BMI. The respective median age was 79, 75.5, and 70 years, which was significantly different between these groups. Severity (APACHE II and SOFA) scores were similar among the study groups.

 **Table 2** shows the comparison between the low BMI and non-low BMI groups.

### Associations between BMI and outcomes

The 28-day all-cause mortality differed among the groups (p = 0.032 in the three-group comparison) (**Table 3**), whereas the in-hospital mortality and ICU-free days were not significantly different (**Table 3**). Patients in the low BMI group had a higher odds ratio (OR) for 28-day all-cause mortality than those in the normal BMI group in both the crude (OR: 2.2, 95% confidence interval [CI]: 1.2–3.9; p = 0.010) and adjusted (OR: 2.4, 95% CI: 1.2–4.6; p = 0.009) analyses (**S1 Table**). There were no significant differences in the ORs for in-hospital mortality among the three study groups in both the crude and adjusted models (**S2 Table**).

### Comparison between the low and non-low BMI groups

The 28-day mortality rates were higher in the low BMI group than in the non-low BMI group (21.4% [22/103] *vs.* 13.9% [54/445]; p = 0.014) (**Table 4**).

 Relative to the non-low BMI group, the crude and adjusted ORs for 28-day mortality in the low BMI group were 2.0 (95% CI: 1.1–3.4) and 2.3 (95% CI: 1.2–4.2), respectively, in the logistic regression model (**Table 5**). Low BMI (hazard ratio [HR] 1.7, 95% CI: 1.0–2.8) was significantly associated with shorter survival duration in the Cox proportional hazards regression

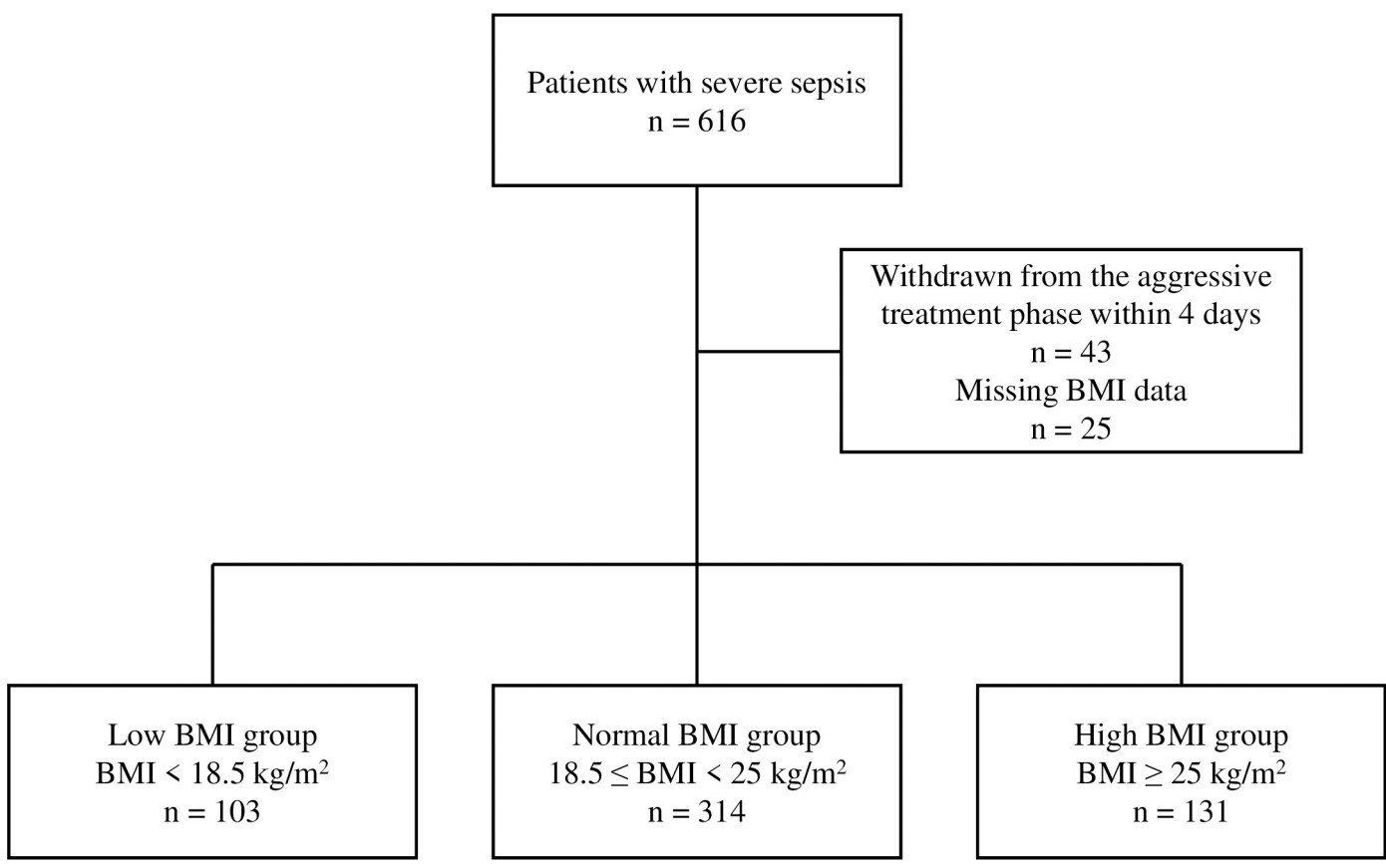

**Fig 1. Flowchart of the patients enrolled in the Tohoku Sepsis Registry who were included in this study.** BMI, body mass index.

model (**Table 6**). **S1 Fig** shows the survival curves comparing the low BMI and non-low BMI groups (p = 0.041, log-rank test). The missing values and our sensitivity analyses were described in **S3 Table**. The in-hospital mortality and median ICU-free days in the low BMI group were not significantly different from those in the non-low BMI group.

## Discussion

In the present study, we found that Japanese patients with sepsis who had a low BMI had a higher 28-day mortality rate than those with a non-low BMI.

Previous studies on ICU patients have focused on outcomes regarding obesity *vs*. non-obesity, neglecting the influence of a low BMI. In a recent epidemiological study in Oceania [29], 2.9% of the enrolled participants accounted for patients with a low BMI with a median age of 60 years. In an observational study of 148,783 ICU patients using data from the eICU Collaborative Research Database—a multicenter ICU database with high granularity data for over 200,000 admissions to ICUs monitored by eICU Programs across the United States (Philips Healthcare, a major vendor of ICU equipment and services, provides a teleICU service known as the eICU program), the composition ratio of patients with a low BMI was 4.3% and the median age was 67 years [30]. In four previous observational studies on Western patients with sepsis, the composition ratios of patients with a low BMI were 2.6%, 6.8%, 6.2%, and 8.6% and the median age was 63.2, 59.1, 60.3, and 57.5 years, respectively [31–34]. In contrast, the proportion of patients with a low BMI in the present study on Japanese patients was 18.8% and

**Table 1. Patient demographics and clinical data of the three study groups defined by the BMI range.**

| Variable | Low BMI group (<18.5 kg/m²) (n = 103) | Normal BMI group (18.5≤ BMI <25.0 kg/m²) (n = 314) | High BMI group (≥25.0 kg/m²) (n = 131) | p-value |
|---|---|---|---|---|
| **BMI (kg/m²), median (IQR)** | 16.8[†,‡](15.7–17.6) | 21.9[‡](20.2–23.1) | 27.5 (26.0–30.0) | <0.001 |
| **Age (years), median (IQR)** | 79.0[‡] (68–86) | 75.5[‡] (64–83) | 70.0 (61–79) | <0.001 |
| **Male, n (%)** | 63 (61.2) | 199 (63.4) | 83 (63.4) | 0.92 |
| **Pre-existing conditions, n (%)** | | | | |
| **Cardiovascular disease** | 12 (11.7) | 42 (13.4) | 19 (14.5) | 0.82 |
| **Stroke** | 17 (16.5) | 48 (15.3) | 12 (9.2) | 0.173 |
| **COPD** | 7 (6.8) | 9 (2.9) | 3 (2.3) | 0.117 |
| **Autoimmune disease** | 4 (3.9) | 17 (5.4) | 8 (6.1) | 0.74 |
| **Chronic liver disease** | 1 (1.0) | 11 (3.5) | 6 (4.6) | 0.29 |
| **Diabetes mellitus** | 21[‡](20.4) | 95 (30.3) | 48 (36.6) | 0.026 |
| **Chronic kidney disease** | 7 (6.8) | 34 (10.8) | 14 (10.7) | 0.48 |
| **Malignancy** | 13 (12.6) | 32 (10.2) | 12 (9.2) | 0.68 |
| **Medications before admission, n (%)** | | | | |
| **Steroids** | 10 (9.7) | 40 (12.7) | 19 (14.5) | 0.54 |
| **Immunosuppressant drugs** | 2 (1.9) | 11 (3.5) | 9 (6.9) | 0.128 |
| **Statins** | 10 (9.7) | 46 (14.6) | 23 (17.6) | 0.23 |
| **Anti-platelets** | 16 (15.5) | 59 (18.8) | 20 (15.3) | 0.56 |
| **β-blockers** | 14 (13.6) | 30 (9.6) | 18 (13.7) | 0.33 |
| **Severity** | | | | |
| **Lactate (mmol/L), median (IQR)** | 2.75 (2.08–4.03) | 2.80 (2.05–4.40) | 2.50 (1.70–4.30) | 0.23 |
| **Septic shock, n (%)** | 60 (58.3) | 158[‡](50.3) | 84 (64.1) | 0.027 |
| **APACHE II score, median (IQR)** | 20 (15–26) | 20(15–26) | 19 (14–27) | 0.83 |
| **SOFA score, median (IQR)** | 7 (5.0–11.0) | 8 (5.0–11.0) | 8 (5.0–11.5) | 0.47 |

Abbreviations: APACHE, Acute Physiology and Chronic Health Evaluation; BMI, body mass index; COPD, chronic obstructive pulmonary disease; IQR, interquartile range; SOFA, Sequential Organ Failure Assessment.

†: vs. normal BMI group, ‡: vs. high BMI group.

† and ‡: p-values <0.016 for multiple comparisons (after Bonferroni correction) were considered statistically significant.

the median age was 79 years. Therefore, the patients' characteristics of this study differ from those of previous studies on Western populations. A recent cohort study in China reported that a low BMI is an independent factor associated with reduced 90-day survival in medical patients with sepsis. In this Chinese report, the proportion of patients with a low BMI was 18.5% and the median age was 79 years [17]. These results are consistent with the findings of our study.

The pathophysiological reasons for poor outcomes in patients with sepsis who have a low BMI are unknown. Gentile *et al*. [35] proposed the concept of persistent inflammation, immunosuppression, and catabolism syndrome that includes a BMI of <18 kg/m². Adipose tissue has various protective effects related to storage functions for catabolic pathways and the tolerance of inflammatory responses [3]. In general, patients with a low BMI are considered to have a reduced amount of adipose tissue [7–9]. Thus, it is possible that its bioprotective effects are reduced. Pepper *et al*. [16] suggest that malnutrition may be associated with poor outcomes in patients with a low BMI. The BMI cutoff value for malnutrition in Asians was recently reported to be 17.0–17.8 kg/m² [36]. Thus, the BMI in persistent inflammation, immunosuppression, and catabolism syndrome and malnutrition is close to the that used to define the low

**Table 2. Patient demographics and clinical data of the low and non-low BMI groups.**

| Variable | Low BMI group (<18.5 kg/m$^2$) (n = 103) | Non-low BMI group (≥18.5 kg/m$^2$) (n = 445) | p-value |
|---|---|---|---|
| **BMI (kg/m$^2$), median (IQR)** | 16.8 (15.7–17.6) | 22.9 (21.0–25.6) | <0.001 |
| **Age (years), median (IQR)** | 79 (68–86) | 73 (64–82) | 0.003 |
| **Male, n (%)** | 63 (61.2) | 282 (63.4) | 0.73 |
| **Pre-existing conditions, n (%)** | | | |
| **Cardiovascular disease** | 12 (11.7) | 61 (13.7) | 0.63 |
| **Stroke** | 17 (16.5) | 60 (13.5) | 0.43 |
| **COPD** | 7 (6.8) | 12 (2.7) | 0.065 |
| **Autoimmune disease** | 4 (3.9) | 25 (5.6) | 0.63 |
| **Chronic liver disease** | 1 (1.0) | 17 (3.8) | 0.22 |
| **Diabetes mellitus** | 21 (20.4) | 143 (32.1) | 0.023 |
| **Chronic kidney disease** | 7 (6.8) | 48 (10.8) | 0.28 |
| **Malignancy** | 13 (12.6) | 44 (9.9) | 0.47 |
| **Medications before admission, n (%)** | | | |
| **Steroids** | 10 (9.7) | 59 (13.3) | 0.41 |
| **Immunosuppressant drugs** | 2 (1.9) | 20 (4.5) | 0.40 |
| **Statins** | 10 (9.7) | 69 (15.6) | 0.161 |
| **Anti-platelets** | 16 (15.5) | 79 (17.8) | 0.67 |
| **β-blockers** | 14 (13.6) | 48 (10.8) | 0.49 |
| **Severity** | | | |
| **Lactate (mmol/L), median (IQR)** | 2.75 (2.08–4.03) | 2.70 (1.90–4.40) | 0.99 |
| **Septic shock, n (%)** | 60 (58.8) | 242 (55.1) | 0.51 |
| **APACHE II score, median (IQR)** | 20 (15–26) | 19 (15–26) | 0.57 |
| **SOFA score, median (IQR)** | 7 (5–11) | 8 (5–11) | 0.34 |

Abbreviations: APACHE, Acute Physiology and Chronic Health Evaluation; BMI, body mass index; COPD, chronic obstructive pulmonary disease; IQR, interquartile range; SOFA, Sequential Organ Failure Assessment.

BMI group in our study. Patients with a low BMI at the time of admission may be inherently more prone to inflammation and more malnourished; these were difficult to assess appropriately in this study.

There are several limitations. First, the retrospective nature of this study was able to demonstrate an association between a low BMI and mortality but did not allow for causal inference. Second, as the Tohoku Sepsis Registry was conducted in 2015, the definition of sepsis differed from the new definition presented in 2016 [1]. Patient selection may have been different if the

**Table 3. BMI and sepsis outcomes.**

| Variable | Low BMI group (<18.5 kg/m$^2$) (n = 103) | Normal BMI group (18.5≤ BMI <25.0 kg/m$^2$) (n = 314) | High BMI group (≥25.0 kg/m$^2$) (n = 131) | p-value |
|---|---|---|---|---|
| **28-day mortality, n (%)** | 22[†] (21.4) | 35 (11.2) | 19 (14.5) | 0.032 |
| **In-hospital mortality, n (%)** | 25 (24.3) | 51 (16.2) | 28 (21.4) | 0.142 |
| **ICU-free days, median (IQR)** | 18 (4–24) | 22 (12–25) | 20 (5–24) | 0.066 |

Abbreviations: BMI, body mass index; ICU, intensive care unit; IQR, interquartile range.

†: vs. normal BMI group.

†: p-values <0.016 for multiple comparisons (after Bonferroni correction) were considered statistically significant.

**Table 4. Outcomes in patients in the low and non-low BMI groups.**

| Variable | Low BMI group (<18.5 kg/m$^2$) (n = 103) | Non-low BMI group (≥18.5 kg/m$^2$) (n = 445) | p-value |
|---|---|---|---|
| 28-day mortality, n (%) | 22 (21.4) | 54 (13.9) | 0.014 |
| In-hospital mortality, n (%) | 25 (24.3) | 79 (17.8) | 0.128 |
| ICU-free days, median (IQR) | 18 (4–24) | 21 (10–25) | 0.072 |

Abbreviations: BMI, body mass index; ICU, intensive care unit; IQR, interquartile range.

**Table 5. Association between BMI and 28-day mortality in patients with sepsis.**

| Variable | OR | 95% CI | p-value |
|---|---|---|---|
| **Crude** | | | |
| Low *vs*. non-low BMI group | 2.0 | 1.1–3.4 | 0.016 |
| **Adjusted** | | | |
| Low *vs*. non-low BMI group | 2.3 | 1.2–4.2 | 0.01 |
| Age | 1.0 | 0.9–1.0 | 0.86 |
| Sex (male vs. female) | 1.7 | 0.9–3.0 | 0.08 |
| APACHE II scores | 1.1 | 1.0–1.1 | 0.006 |
| SOFA scores | 1.1 | 0.9–1.2 | 0.07 |
| Shock | 1.0 | 0.4–2.0 | 0.90 |
| Pre-existing conditions | 1.0 | 0.5–1.8 | 0.95 |
| Lactate level | 1.1 | 0.9–1.1 | 0.15 |

Abbreviations: APACHE, Acute Physiology and Chronic Health Evaluation; BMI, body mass index; CI, confidence interval; OR, odds ratio; SOFA, Sequential Organ Failure Assessment.

**Table 6. Cox proportional hazards regression model for 28-day mortality.**

| Variable | HR | 95% CI | p-value |
|---|---|---|---|
| Low *vs*. non-low BMI group | 1.7 | 1.0–2.8 | 0.042 |
| Age | 1.0 | 0.9–1.0 | 0.36 |
| Sex (male vs. female) | 1.5 | 0.8–2.5 | 0.14 |
| APACHE II scores | 1.0 | 1.0–1.0 | 0.026 |
| SOFA scores | 1.1 | 0.9–1.2 | 0.060 |
| Shock | 0.7 | 0.3–1.4 | 0.34 |
| Pre-existing conditions | 0.9 | 0.5–1.4 | 0.61 |
| Lactate level | 1.1 | 1.0–1.1 | 0.017 |

Abbreviations: APACHE, Acute Physiology and Chronic Health Evaluation; BMI, body mass index; CI, confidence interval; OR, odds ratio; SOFA, Sequential Organ Failure Assessment.

new sepsis definition had been applied in 2015. Third, the cutoff points based on BMI differs between global populations, such as South Asians, Chinese, Aboriginals, and Europeans [37]. We used the Japanese guidelines in this study [26]. Different results may have been obtained if other definitions had been applied. However, it is assumed that the Japanese guidelines reflect the characteristics of our study population the most [19, 26]. Finally, the variables used for

adjustment in the multivariate analysis were limited because the number of patients and non-survivors was not large enough. Given that there were 76 non-survivors based on the number of event occurrences, we used eight adjustment variables to develop a suitable statistical model. These findings may be useful for risk stratification and resource distribution in an ICU setting. In the future, larger cohort studies may reveal whether the association between low BMI and poor outcomes in patients with sepsis is also observed in other countries or in patients of other ethnicities.

## Conclusions

In conclusion, this study showed that a low BMI was associated with increased 28-day mortality and may be a risk factor for poor outcomes in Japan. Japanese older adults often have a low BMI, representing a population at an increased risk of negative outcomes related to sepsis.

## Supporting information

**S1 Fig. The 28-day survival curves for patients with sepsis in the low BMI and non-low BMI groups.** BMI, body mass index.
(TIF)

**S1 Table. Association between BMI and 28-day mortality in patients with sepsis.** BMI: body mass index.
(DOCX)

**S2 Table. Association between BMI and in-hospital mortality in patients with sepsis.** BMI: body mass index.
(DOCX)

**S3 Table. The proportion of missing values in each group and sensitivity analyses.**
(DOCX)

## Acknowledgments

We would like to acknowledge all the Tohoku Sepsis Registry investigators who contributed to the collection and assessment of the data at each institution. The authors are grateful to S. Osaki for data management and administrative support. This manuscript was edited by a native English speaker associated with Editage, Tokyo, Japan.

## Author Contributions

**Conceptualization:** Tetsuya Sato, Daisuke Kudo, Shigeki Kushimoto.

**Data curation:** Daisuke Kudo, Masatsugu Hasegawa, Fumihito Ito, Sathoshi Yamanouchi, Hiroyuki Honda, Kohkichi Andoh, Hajime Furukawa, Yasuo Yamada, Yuta Tsujimoto, Manabu Okuyama, Masakazu Kobayashi.

**Formal analysis:** Tetsuya Sato, Masakazu Kobayashi.

**Investigation:** Daisuke Kudo, Masatsugu Hasegawa, Fumihito Ito, Sathoshi Yamanouchi, Hiroyuki Honda, Kohkichi Andoh, Hajime Furukawa, Yasuo Yamada, Yuta Tsujimoto, Manabu Okuyama.

**Methodology:** Tetsuya Sato, Daisuke Kudo, Shigeki Kushimoto, Masakazu Kobayashi.

**Writing – original draft:** Tetsuya Sato.

**Writing – review & editing:** Tetsuya Sato, Daisuke Kudo, Shigeki Kushimoto, Masatsugu Hasegawa, Fumihito Ito, Sathoshi Yamanouchi, Hiroyuki Honda, Kohkichi Andoh, Hajime Furukawa, Yasuo Yamada, Yuta Tsujimoto, Manabu Okuyama, Masakazu Kobayashi.

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
