## [Decision Letter · Decision Letter 0]

5 Feb 2021

PONE-D-20-41042

Associations between low body mass index and mortality in patients with sepsis: A retrospective analysis of a cohort study in Japan

PLOS ONE

Dear Dr. Sato,

Thank you for submitting your manuscript to PLOS ONE. After careful consideration, we feel that it has merit but does not fully meet PLOS ONE’s publication criteria as it currently stands. Therefore, we invite you to submit a revised version of the manuscript that addresses the points raised during the review process.

Thank you very much for submitting the study which is reported association between BMI and mortality in patients with sepsis . Overall, well written and include attractive aspects of the study. However, some issues have raised and respond to our reviewers comments. 

We look forward to receiving your revised manuscript.

Kind regards,

Yutaka Kondo

Academic Editor

PLOS ONE

2. In the ethics statement in the manuscript and in the online submission form, please provide additional information about the patient records/samples used in your retrospective study, including: a) whether all data were fully anonymized before you accessed them; b) the date range (month and year) during which patients' medical records/samples were accessed.

3. Thank you for including your ethics statement:  "This research was approved by the Ethics Review Board based on the guidelines of the Ethics Committee of the Graduate School of Medicine, Tohoku University (No.2013-1-42). And the Institutional Review Boards (IRB) of each of the 10 participating institutions approved the study. According to the Japanese Guideline (Ministry of Education, Culture, Sports, Science and Technology, and Ministry of Health, Labor and Welfare, Japan. Ethical Guidelines for Medical and Health Research Involving Human Subjects. March 2015), all the IRBs waived the need for informed consent due to the observational study design and there being no treatment beyond the daily clinical practice.".   

Reviewers' comments:

Reviewer's Responses to Questions

**Comments to the Author**

1. Is the manuscript technically sound, and do the data support the conclusions?

Reviewer #1: Yes

Reviewer #2: Yes

2. Has the statistical analysis been performed appropriately and rigorously? 

Reviewer #1: N/A

Reviewer #2: Yes

3. Have the authors made all data underlying the findings in their manuscript fully available?

Reviewer #1: Yes

Reviewer #2: Yes

4. Is the manuscript presented in an intelligible fashion and written in standard English?

Reviewer #1: Yes

Reviewer #2: Yes

5. Review Comments to the Author

Reviewer #1: Thank you for giving me the opportunity to review this manuscript. The manuscript relates to the associations between low body mass index and mortality in patients with sepsis. This work is essential for reveal the effect of malnutrition or low BMI for sepsis. I have some concerns about the manuscript in its present form, which I detail herein:

1.You should reconstruct the abstract as a structured abstract(i.e., Background, method, result, conclusion).

2.In figure 1, you should mention what statistical analysis applied for obtaining the p-value. In addition, you should use "*,† or ‡" for showing the significance instead of "a" or "b"

3.Why you chose the logistic model to evaluate 28-days mortality even though Wacharasint et al (Wacharasint et al, Crit Care, 2013) used survival analysis for this issue? Don't you use cox proportional hazard model for this data?

4.You should show the sample size calculation in the method part.

5.In table 5, you showed the important result, but I would like to know the whole result of the adjusted model. ( age, sex, APACHE IIscores, SOFAscores, shock, and pre-existing conditions.)

6.Page 15 line 208 mistype "eICU".

Reviewer #2: Page 5, Section "Materials and methods," subsection "Study design": The authors are requested to briefly describe Tohoku Sepsis Registry, especially regarding how data is collected and whether it is extracted from medical records/billing records/patient surveys.

Page 6, Section "Materials and methods," subsection "Definitions and outcome measures": The authors have categorized BMI into low, normal, and high based on Japanese Guidelines for Management of Obesity Disease. Did the authors consider keeping BMI as a continuous variable and analyzing the association of BMI (as a continuous variable) with 28-day mortality and the secondary outcomes (all-cause in-hospital mortality and ICU-free days)?

Page 6, Section "Materials and methods," subsection "Definitions and outcome measures": The authors are requested to describe how the 28-day mortality was measured? Were the patients followed up after discharge? Was the mortality all-cause mortality?

Page 6, Section "Materials and methods," subsection "Statistical analyses":

• The authors mention that the dataset had missing values. The authors are requested to describe how much of the data was missing and whether it affected the results from the analyses.

• How were the pre-existing conditions identified? Were they extracted from medical records or billing codes, or patient interviews?

• The authors are requested whether they considered controlling the multivariable logistic regression for serum lactate levels? It is known that reduction in lactate level is associated with survival from sepsis. Hence including this variable in the model appears to be necessary.

• The authors are requested whether they considered controlling the multivariable logistic regression for other laboratory values, medications, socioeconomic status, etc.?

Page 12, Section "Results," subsection "Associations between BMI and outcomes": The authors are requested to rephrase or remove the following sentence “Although there were no significant differences, patients in the low BMI group had a higher odds ratio…..”. The confidence intervals for the odds ratios mentioned in this sentence are wide; hence it will give the readers a false impression that low BMI had a higher odds ratio for 28-day mortality than high BMI.

Table 5, S1 Table, S2 Table: The authors are requested to provide the odds ratio, 95% CI, and p-values for all the variables (such as age, sex, APACHE II scores, etc.) in the regression model.

Page 16, Section "Discussion": The authors are requested to explain what they mean by “confounders used for adjustment in the multivariate analysis were limited because the number of patients and non-survivors was not large enough.”

6. PLOS authors have the option to publish the peer review history of their article (what does this mean?). If published, this will include your full peer review and any attached files.

Reviewer #1: **Yes: **Yujiro Matshishi

Reviewer #2: No

---

## [Author Response · Author response to Decision Letter 0]

19 Mar 2021

Responses to Reviewers

Reviewer #1:

Q1. You should reconstruct the abstract as a structured abstract (i.e., Background, method, result, conclusion).

A1. Thank you for your suggestion. We agree that this format would be more appropriate. Accordingly, we have included the required subheadings in the revised Abstract (p. 3, lines 29-49, “Abstract”).

Q2. In figure 1, you should mention what statistical analysis applied for obtaining the p-value. In addition, you should use "*, † or ‡" for showing the significance instead of "a" or "b"

A2. Thank you for this suggestion. In accordance with your recommendation, we have revised Table 1 to include “†” and “‡” to indicate statistical significance. In addition, the following is included in the legend below the table: “†　and　‡: p values < 0.016 for multiple comparisons (after Bonferroni correction) were considered statistically significant” (p. 8-10, Table 1). A similar statement is included in the Statistical analysis subsection of the Materials and methods (p. 7-8, lines 143-145).

Q3. Why you chose the logistic model to evaluate 28-days mortality even though Wacharasint et al (Wacharasint et al, Crit Care, 2013) used survival analysis for this issue? Don't you use cox proportional hazard model for this data?

A3. We thank the reviewer for expressing this concern. As suggested, we utilized a Cox proportional hazards model for survival analysis. The relevant text was added in the Materials and methods (p. 7, lines 131-132) and Results sections (p. 14, lines 205-208, p. 15-16, Table 6). The Kaplan–Meier curves are also shown in S1 Figure (p. 25, lines 388-389).

Table 6. Cox proportional hazards regression model for 28-day mortality

Variable HR 95% CI p-value

Low vs. non-low BMI group 1.7 1.0–2.8 0.042

Age 1.0 0.9–1.0 0.36

Sex (male vs. female) 1.5 0.8–2.5 0.14

APACHE Ⅱ scores 1.0 1.0–1.0 0.026

SOFA scores 1.1 0.9–1.2 0.060

Shock 0.7 0.3–1.4 0.34

Pre-existing conditions 0.9 0.5–1.4 0.61

Lactate level 1.1 1.0–1.1 0.017

Abbreviations: APACHE, Acute Physiology and Chronic Health Evaluation; BMI, body mass index; CI, confidence interval; OR, odds ratio; SOFA, Sequential Organ Failure Assessment.

S1 Figure

Q4. You should show the sample size calculation in the method part.

A4. You have raised an important point; however, unfortunately, this study was a secondary analysis that was not planned in advance, and there was no preset sample size. This point is now described in the Materials and methods section (p. 7, lines 140-141).

Q5. In table 5, you showed the important result, but I would like to know the whole result of the adjusted model. (age, sex, APACHE II scores, SOFA scores, shock, and pre-existing conditions.)

A5. Thank you for your suggestion. We agree that the complete results should be included and have thus revised Table 5 in accordance with your recommendation (p. 14-15, Table 5). 

Table 5. Association between BMI and 28-day mortality in patients with sepsis

Variable OR 95% CI p-value

Crude 

 Low vs. non-low BMI group 2.0 1.1–3.4 0.016

Adjusteda 

 Low vs. non-low BMI group 2.3 1.2–4.2 0.01

Age 1.0 0.9–1.0 0.86

Sex (male vs. female) 1.7 0.9–3.0 0.08

APACHE II scores 1.1 1.0–1.1 0.006

SOFA scores 1.1 0.9–1.2 0.07

Shock 1.0 0.4–2.0 0.90

Pre-existing conditions 1.0 0.5–1.8 0.95

Lactate level 1.1 0.9–1.1 0.15

Abbreviations: APACHE, Acute Physiology and Chronic Health Evaluation; BMI, body mass index; CI, confidence interval; OR, odds ratio; SOFA, Sequential Organ Failure Assessment.

Q6. Page 15, line 208 mistype "eICU".

A6. Thank you for your comment. We have clarified that “the eICU Collaborative Research Database” refers to a multicenter ICU database that contains high granularity data for over 200,000 admissions to ICUs monitored by eICU Programs across the United States (Philips Healthcare, a major vendor of ICU equipment and services, provides a teleICU service known as the eICU program) (p. 16, lines 227-231).

Reviewer #2: 

Q1. Page 5, Section "Materials and methods," subsection "Study design": The authors are requested to briefly describe Tohoku Sepsis Registry, especially regarding how data is collected and whether it is extracted from medical records/billing records/patient surveys.

A1. Thank you for your suggestion. In accordance with your recommendation, we have revised the manuscript to include the following sentences:

 “This study used data from the Tohoku Sepsis Registry (University Hospital Medical Information Network Clinical Trials Registry: UMIN000010297), a multicenter observational cohort study conducted at 10 institutions—including three university hospitals and seven community hospitals—in the Tohoku District in northern Japan.

 The detailed design of this study has been reported previously [22]. Briefly, the Tohoku Sepsis Registry prospectively enrolled consecutive patients admitted to an intensive care unit (ICU) with severe sepsis or patients who developed severe sepsis after admission to an ICU between January 2015 and December 2015. Severe sepsis and septic shock were defined according to the 2012 International Sepsis Guidelines [23]. Patients aged <18 years were excluded from the registry. Researchers at each institution collected data from patient medical records and registered them in the web registration system.” (p. 5-6, lines 92-102).

Q2. Page 6, Section "Materials and methods," subsection "Definitions and outcome measures": The authors have categorized BMI into low, normal, and high based on Japanese Guidelines for Management of Obesity Disease. Did the authors consider keeping BMI as a continuous variable and analyzing the association of BMI (as a continuous variable) with 28-day mortality and the secondary outcomes (all-cause in-hospital mortality and ICU-free days)?

A2. Thank you for your suggestion. You have raised an important question. However, continuous variables are commonly used when the relationship between variables and outcomes are predicted to exhibit linear regression. We have searched the literature for previous studies regarding whether linear or non-linear regression is predicted for the relationship between BMI and the outcome investigated in the present study. Based on the reports by Zhang et al. [1], Sakr et al. [2], and Niedziela et al. [3], non-linear regression was predicted for the relationship between BMI and mortality (please see the figures in each of the references below). Thus, in our case, we judged it appropriate to use BMI as a categorical variable. In fact, the crude 28-day mortality rates in this study were 21.4% in the low BMI group, 11.2% in the normal BMI group, and 14.5% in the high BMI group, indicating a U-shaped relationship between BMI and mortality (i.e., non-linear regression). The same tendency was observed for crude in-hospital mortality, although the relationship between BMI and ICU-free days exhibited a reversed U-shape.

1. Zheng, W., et al. Association between body-mass index and risk of death in more than 1 million Asians. N Engl J Med. 2011; 364(8): 719-729.

2. Sakr, Y., et al. Being overweight or obese is associated with decreased mortality in critically ill patients: a retrospective analysis of a large regional Italian multicenter cohort. J Crit Care. 2012; 27(6): 714-721.

3. Niedziela, J., et al. The obesity paradox in acute coronary syndrome: a meta-analysis. Eur J Epidemiol. 2014; 29(11): 801-812.

Q3. Page 6, Section "Materials and methods," subsection "Definitions and outcome measures": The authors are requested to describe how the 28-day mortality was measured? Were the patients followed up after discharge? Was the mortality all-cause mortality?

A3. Thank you for your suggestion. As indicated, we have revised the text as follows: “The primary outcome measure was all-cause 28-day mortality. No follow-up after discharge, survival discharge, or survival during hospitalization within 28 days was regarded as survival, and in-hospital death within 28 days was regarded as death.” (p. 6-7, lines 118-120).

Q4. Page 6, Section "Materials and methods," subsection "Statistical analyses":

Q4-1. The authors mention that the dataset had missing values. The authors are requested to describe how much of the data was missing and whether it affected the results from the analyses.

A4-1. Thank you for your suggestion. In accordance with your recommendation, we have described the missing values and sensitivity analyses in S3 Table (p. 14, lines 208-209, p. 25, line 394). In addition, to account for the signiﬁcant proportions of missing values for APACHE II (9.9%) and SOFA scores (9.3%) that were assumed to be missing at random, we conducted our sensitivity analyses with multiple imputation. Multiple imputation through chained equations with predictive mean matching was employed to impute all missing values for the variables and outcomes in the dataset for the logistic regression model. Multiple imputation generated 20 datasets with 20 iterations. As a result, relative to the non-low BMI group, the adjusted OR for 28-day mortality in the low BMI group was 2.3 (95% CI: 1.2–4.2) (S3 Table). This result was consistent with the results of the analysis of the original data.

S3 Table. The proportion of missing values in each group

Variable Low BMI group

n, (%) Non-low BMI group

n, (%)

Age 0 0

Sex 0 0

APACHE Ⅱ scores 6 (5.8) 48 (10.8)

SOFA scores 5 (4.9) 46 (10.3)

Shock 1 (1) 6 (1.3)

Pre-existing conditions 0 0

Lactate 1 (1) 7 (1.6)

Abbreviations: APACHE, Acute Physiology and Chronic Health Evaluation; BMI, body mass index; SOFA, Sequential Organ Failure Assessment.

Q4-2. How were the pre-existing conditions identified? Were they extracted from medical records or billing codes, or patient interviews?

A4-2. Thank you for these relevant questions. Please see A1 above. The revised text now indicates that researchers at each institution retrieved the data regarding pre-existing conditions from patient medical records (p. 6, lines 101-102).

Q4-3. The authors are requested whether they considered controlling the multivariable logistic regression for serum lactate levels? It is known that reduction in lactate level is associated with survival from sepsis. Hence including this variable in the model appears to be necessary.

A4-3. Thank you for your suggestion. We agree that this information is necessary and have incorporated data for lactate levels in the text and tables (p. 7, line 136, p. 14-16, Tables 5 and 6).

Table 5. Association between BMI and 28-day mortality in patients with sepsis

Variable OR 95% CI p-value

Crude 

 Low vs. non-low BMI group 2.0 1.1–3.4 0.016

Adjusted 

 Low vs. non-low BMI group 2.3 1.2–4.2 0.01

Age 1.0 0.9–1.0 0.86

Sex (male vs. female) 1.7 0.9–3.0 0.08

APACHE II scores 1.1 1.0–1.1 0.006

SOFA scores 1.1 0.9–1.2 0.07

Shock 1.0 0.4–2.0 0.90

Pre-existing conditions 1.0 0.5–1.8 0.95

Lactate level 1.1 0.9–1.1 0.15

Abbreviations: APACHE, Acute Physiology and Chronic Health Evaluation; BMI, body mass index; CI, confidence interval; OR, odds ratio; SOFA, Sequential Organ Failure Assessment.

Table 6. Cox proportional hazards regression model for 28-day mortality

Variable HR 95% CI p-value

Low vs. non-low BMI group 1.7 1.0–2.8 0.042

Age 1.0 0.9–1.0 0.36

Sex (male vs. female) 1.5 0.8–2.5 0.14

APACHE Ⅱ scores 1.0 1.0–1.0 0.026

SOFA scores 1.1 0.9–1.2 0.060

Shock 0.7 0.3–1.4 0.34

Pre-existing conditions 0.9 0.5–1.4 0.61

Lactate 1.1 1.0–1.1 0.017

Abbreviations: APACHE, Acute Physiology and Chronic Health Evaluation; BMI, body mass index; CI, confidence interval; OR, odds ratio; SOFA, Sequential Organ Failure Assessment.

Q4-4. The authors are requested whether they considered controlling the multivariable logistic regression for other laboratory values, medications, socioeconomic status, etc.?

A4-4. Thank you for your suggestion. You have raised an important point; however, there was a limit to the number of variables that could be adopted based on the number of outcomes, and variables considered to be clinically more important were adopted. Laboratory values and medications are also important, but they are judged to have low priority. To address this issue, we have revised the text as follows: “We deemed these variables potentially important based on clinical judgment and past sepsis research [27]. A previous study also indicated that body weight may be associated with age and pre-existing conditions [28].” (p. 7, lines 136-138). There were no data related to socioeconomic status. Please also see A7 below.

Q5. Page 12, Section "Results," subsection "Associations between BMI and outcomes": The authors are requested to rephrase or remove the following sentence “Although there were no significant differences, patients in the low BMI group had a higher odds ratio…..”. The confidence intervals for the odds ratios mentioned in this sentence are wide; hence it will give the readers a false impression that low BMI had a higher odds ratio for 28-day mortality than high BMI.

A5. Thank you for this suggestion. As indicated, we have deleted the following sentence: “Although there were no significant differences, patients in the low BMI group had a higher odds ratio…” (p. 12, line 186).

Q6. Table 5, S1 Table, S2 Table: The authors are requested to provide the odds ratio, 95% CI, and p-values for all the variables (such as age, sex, APACHE II scores, etc.) in the regression model.

A6. Thank you for this suggestion. We agree that information regarding these variables should be included in the tables. Accordingly, we have incorporated your suggestion in Table 5 (p. 14-15), S1 Table, and S2 Table (p. 25, lines 390-393).

Table 5 (see above)

S1 Table. Association between BMI and 28-day mortality in patients with sepsis

Variable OR (95% CI) p-value

Crude 

Low vs. normal BMI group 2.2 (1.2–3.9) 0.010

High vs. normal BMI group 1.4 (0.7–2.5) 0.32

Low vs. high BMI group 1.6 (0.8–3.2) 0.173

Adjusted 

Low vs. normal BMI group 2.4 (1.2–4.7) 0.009

High vs. normal BMI group 1.3 (0.6–2.5) 0.53

Low vs. high BMI group 1.9 (0.9–4.3) 0.094

Age 1.0 (0.9–1.0) 0.83

Sex (male vs. female) 1.7 (0.9–3.0) 0.09

APACHE II scores 1.1 (1.0–1.1) 0.006

SOFA scores 1.1 (0.9–1.2) 0.07

Shock 0.9 (0.4–1.9) 0.85

Pre-existing conditions 1.0 (0.5–1.8) 0.98

Lactate level 1.1 (0.9–1.1) 0.14

Abbreviations: APACHE, Acute Physiology and Chronic Health Evaluation; BMI, body mass index; CI, confidence interval; OR, odds ratio; SOFA, Sequential Organ Failure Assessment.

Table S2. Association between BMI and in-hospital mortality in patients with sepsis

Variable OR (95% CI) p-value

Crude 

Low vs. normal BMI group 1.7 (0.96–2.8) 0.068

High vs. normal BMI group 1.4 (0.8–2.3) 0.197

Low vs. high BMI group 1.2 (0.6–2.2) 0.59

Adjusted 

Low vs. normal BMI group 1.7 (0.95–3.2) 0.070

High vs. normal BMI group 1.3 (0.7–2.2) 0.42

Low vs. high BMI group 1.4 (0.6–2.7) 0.37

Age 1.0 (0.9–1.0) 0.50

Sex (male vs. female) 1.8 (1.0–3.0) 0.026

APACHE II scores 1.1 (1.0–1.1) 0.003

SOFA scores 1.1 (0.9–1.1) 0.16

Shock 1.4 (0.7–2.7) 0.26

Pre-existing conditions 1.1 (0.6–1.8) 0.80

Lactate level 1.0 (0.9–1.1) 0.41

Q7. Page 16, Section "Discussion": The authors are requested to explain what they mean by “confounders used for adjustment in the multivariate analysis were limited because the number of patients and non-survivors was not large enough.”

A7. Thank you for your suggestion. We have replaced the term [confounders] throughout the paper with [variables] to use more precise terms. We have also revised the text to include the following: “Given that there were 76 non-survivors based on the number of event occurrences, we used eight adjustment variables to develop a suitable statistical model.” (p. 18, lines 265-267).

Figures and Tables:

To improve the manuscript based on the reviewers’ comments, we have made the following changes to the figures and tables:

A new figure has been inserted as S1 Figure in the Supporting Information.

Tables 1, 3, and 5 have been revised.

A new table has been inserted as Table 6 in the manuscript.

A new table has been inserted as S3 Table in the Supporting Information.

---

## [Decision Letter · Decision Letter 1]

14 Apr 2021

PONE-D-20-41042R1

Associations between low body mass index and mortality in patients with sepsis: A retrospective analysis of a cohort study in Japan

PLOS ONE

Dear Dr. Sato,

Thank you for submitting your manuscript to PLOS ONE. After careful consideration, we feel that it has merit but does not fully meet PLOS ONE’s publication criteria as it currently stands. Therefore, we invite you to submit a revised version of the manuscript that addresses the points raised during the review process.

ACADEMIC EDITOR: Thank you for revising the manuscript. Please see our reviewers comments.

We look forward to receiving your revised manuscript.

Kind regards,

Yutaka Kondo

Academic Editor

PLOS ONE

Journal Requirements:

Reviewers' comments:

Reviewer's Responses to Questions

**Comments to the Author**

1. If the authors have adequately addressed your comments raised in a previous round of review and you feel that this manuscript is now acceptable for publication, you may indicate that here to bypass the “Comments to the Author” section, enter your conflict of interest statement in the “Confidential to Editor” section, and submit your "Accept" recommendation.

Reviewer #1: All comments have been addressed

Reviewer #2: All comments have been addressed

2. Is the manuscript technically sound, and do the data support the conclusions?

Reviewer #1: Yes

Reviewer #2: Yes

3. Has the statistical analysis been performed appropriately and rigorously? 

Reviewer #1: Yes

Reviewer #2: Yes

4. Have the authors made all data underlying the findings in their manuscript fully available?

Reviewer #1: Yes

Reviewer #2: Yes

5. Is the manuscript presented in an intelligible fashion and written in standard English?

Reviewer #1: Yes

Reviewer #2: Yes

6. Review Comments to the Author

Reviewer #1: Thank you for giving me the opportunity to review this manuscript again. The overall concern was the statistical methodology for the main outcome. This manuscript has the novelty, and now the manuscript is very clearly stated the meaning of this study.

Reviewer #2: Thank you for editing your manuscript as per reviewer suggestions.

Section ‘Material and Methods’: Since the authors have conducted survival analysis, the authors are requested to define how survival was measured. The authors have already provided an explanation for how they measured all-cause 28-day mortality. The authors are requested to add a sentence in this context on how survival was measured such as “survival was measured as number of days from admit date to death” etc. Also, the authors are requested to add a sentence on how patients were censored.

Table 1: Few cells in this table have superscripted ‘b’ instead of † or ‡. The authors are requested to correct them.

7. PLOS authors have the option to publish the peer review history of their article (what does this mean?). If published, this will include your full peer review and any attached files.

Reviewer #1: **Yes: **Yujiro Matsuishi

Reviewer #2: No

---

## [Author Response · Author response to Decision Letter 1]

26 Apr 2021

Responses to Reviewers

Reviewer #1: Thank you for your review of our paper. 

Reviewer #2: Thank you for your comments. Our answers to your points are as follows.

Q1. Section ‘Material and Methods’: Since the authors have conducted survival analysis, the authors are requested to define how survival was measured. The authors have already provided an explanation for how they measured all-cause 28-day mortality. The authors are requested to add a sentence in this context on how survival was measured such as “survival was measured as number of days from admit date to death” etc. Also, the authors are requested to add a sentence on how patients were censored.

A1. Thank you for your suggestion. In accordance with your recommendation, we have revised the manuscript to include the following sentences:

“In survival time analysis, survival was measured as number of days from admit date to death and censor was defined as survival discharge within 28 days” (p. 7, lines 120-122).

Q2. Table 1: Few cells in this table have superscripted ‘b’ instead of † or ‡. The authors are requested to correct them.

A2. Thank you for your point-out, we have corrected Table 1. We have changed ‘b’ to ‘‡’ (p. 9-10, Table 1).

---

## [Decision Letter · Decision Letter 2]

26 May 2021

Associations between low body mass index and mortality in patients with sepsis: A retrospective analysis of a cohort study in Japan

PONE-D-20-41042R2

Dear Dr. Sato,

We’re pleased to inform you that your manuscript has been judged scientifically suitable for publication and will be formally accepted for publication once it meets all outstanding technical requirements.

Kind regards,

Yutaka Kondo

Academic Editor

PLOS ONE

Additional Editor Comments (optional):

Reviewers' comments:

Reviewer's Responses to Questions

**Comments to the Author**

1. If the authors have adequately addressed your comments raised in a previous round of review and you feel that this manuscript is now acceptable for publication, you may indicate that here to bypass the “Comments to the Author” section, enter your conflict of interest statement in the “Confidential to Editor” section, and submit your "Accept" recommendation.

Reviewer #1: All comments have been addressed

Reviewer #2: All comments have been addressed

2. Is the manuscript technically sound, and do the data support the conclusions?

Reviewer #1: Yes

Reviewer #2: Yes

3. Has the statistical analysis been performed appropriately and rigorously? 

Reviewer #1: Yes

Reviewer #2: Yes

4. Have the authors made all data underlying the findings in their manuscript fully available?

Reviewer #1: Yes

Reviewer #2: Yes

5. Is the manuscript presented in an intelligible fashion and written in standard English?

Reviewer #1: Yes

Reviewer #2: Yes

6. Review Comments to the Author

Reviewer #1: Thank you for giving me the opportunity to review this manuscript.

This manuscript is well written, and I accepted this manuscript.

Reviewer #2: (No Response)

7. PLOS authors have the option to publish the peer review history of their article (what does this mean?). If published, this will include your full peer review and any attached files.

Reviewer #1: **Yes: **Yujiro Matsuishi

Reviewer #2: No

---

## [Editor Report · Acceptance letter]

31 May 2021

PONE-D-20-41042R2 

Associations between low body mass index and mortality in patients with sepsis: A retrospective analysis of a cohort study in Japan 

Dear Dr. Sato:

I'm pleased to inform you that your manuscript has been deemed suitable for publication in PLOS ONE. Congratulations! Your manuscript is now with our production department. 

Kind regards, 

on behalf of

Dr. Yutaka Kondo 

Academic Editor

PLOS ONE